# Model Development of A Synergistic Sustainable Marine Ecotourism—A Case Study in Pangandaran Region, West Java Province, Indonesia

**Atikah Nurhayati [1,*], Isah Aisah [2,*] and Asep K. Supriatna [2,*]**

1   Faculty of Fisheries and Marine Science, Padjadjaran University, 45363 West Java, Indonesia
2   Department of Mathematics, Padjadajaran University, 45363 West Java, Indonesia
*   Correspondence: atikah.nurhayati@unpad.ac.id (A.N.); isah.aisah@unpad.ac.id (I.A.);
    a.k.supriatna@unpad.ac.id (A.K.S.); Tel.: +62-08-12-2031417 (A.N.)

**Abstract:** Coastal areas in the South Coast of West Java Province, Indonesia, have potential to develop marine ecotourism. One specific case is the Pangandaran area which must be transferred into economic value by not damaging natural resources. Marine ecotourism development is not only intended to raise foreign exchange for the local government, but is also expected to play a role in maintaining natural resources sustainably. This research aims to analyze the sustainable synergistic marine ecotourism development model. The method used in this research is the quantitative descriptive method. The quantitative descriptive method is used to describe the general condition of the research area, using primary and secondary data. The technique includes the taking of respondents using accidental sampling as many as 50 respondents, consisting of tourists, public figures, and fishermen who have side jobs as providers of marine ecotourism services. The analysis is carried out through the Rapfish modeling approach to measure the synergistic elements of sustainable development of marine ecotourism. Based on the results of the research the ecological dimension of environmental services are the most influential conditions, the economic dimension of marine ecotourism is a less influential condition. Meanwhile, marine ecotourism technology and the social dimension of marine ecotourism are least influential conditions. In regard to infrastructure and regulatory dimensions, the use of information technology is recommended to promote marine ecotourism optimally. It is also concluded that regulations are needed to establish marine ecotourism zoning rules and infrastructure improvements.

**Keywords:** marine ecotourism; coastal areas; fishermen; development model; sustainable

## 1. Introduction

Unsustainable natural resource management practices are an increasing problem in Pangandaran. As overfishing and deforestation continues to degrade the environment, some community members (including fishermen) are looking towards marine ecotourism as a sustainable livelihood alternative. Tourism is a sector made up of many subcategories, such as nature tourism, agrotourism, marine ecotourism, and more. Nature tourism is also called 'ecotourism' which was first conceived by Hector Ceballos-Lascurain in the early 1980 [1]. Tourism is a travel activity in the country and abroad to enjoy natural scenery, such as mountains (plants, wild animals), coastal areas (sea), and cultural aspects of the area. Ecotourism can be realized through an educational approach about natural beauty that can be enjoyed by every tourist [2]. Some considerations in realizing ecotourism are the focus of marine ecotourism in coastal areas, because coastal area is a huge marine tourism asset which is supported by geological potential and characteristics that are very closely related to coral reefs, especially hard

corals, so it is very desirable for development for marine ecotourism such as diving and snorkeling. Ecotourism can contribute to maintaining biodiversity and ecosystem functions [3,4]. Maritime tourism potential of natural resources can be seen in various forms such as coral reef ecosystems, reef fish, ornamental fish, seagrass, and fishing.

In general, the environment is greatly influenced by human activities. The pressure of human activities on natural resources in coastal areas and small islands will have an impact on ecological sustainability [5,6]. The implications of developing marine tourism activities will have an influence on the biotic and abiotic—social, cultural, and economic—environments. Therefore, special considerations are needed to emphasize in the development of marine ecotourism activities. This is due to the fact that marine ecotourism has the potential to cause changes in community behavior, waning social values and norms, loss of identity, as well as social conflict, shifting livelihoods and environmental pollution.

Coastal areas in the South Coast of West Java Province are potential areas for the development of marine ecotourism, one of which is the Pangandaran area. The area could be transferred into an area that produces economic value by not damaging natural resources. Marine ecotourism is one of the two legal income activities in Pangandaran, the other being regulated fishing. Marine ecotourism is a growing sector in Pangandaran and globally. The development of marine ecotourism in the coastal areas of the Pangandaran area will have an effect on people's lives directly or indirectly, especially for fishermen in the Pangandaran region [7].

The development of marine ecotourism in coastal areas will directly involve coastal communities, most of whom work as fishermen. The social characteristics possessed by fishing communities differ from other communities in general. This is caused by differences in the characteristics of the resources concerned [8]. The development of maritime ecotourism is not only intended to increase foreign exchange for local governments, but is expected to play a role as a national scale development building. This is among the reasons why we need to undertake research on synergistic sustainable marine ecotourism.

Maritime ecotourism development has several advantages, namely diversification of work for fishermen, increasing employment opportunities for fishing families, increasing local tax revenues, accelerating the process of income distribution, increasing the added value of ecotourism products, expanding domestic product markets, and providing a multiplier effect on the regional economy [9]. Marine ecotourism development is not only intended to raise foreign exchange for local governments, it is also expected to play a role in maintaining natural resources sustainably. This research aims to analyze the development model of synergistic sustainable marine ecotourism (case study in Pangandaran Region, West Java Province).

## 2. Materials and Methods

This research was conducted from February 2017 to March 2018, taking place in Pangandaran Region (Figure 1). This location has a tropical climate with two seasons, namely the dry season (east season) and the rainy season (west season). The east and west seasons will directly affect the number of visitors in Pangandaran, both domestic and foreign tourists. The east season occurs from May to October, where during this season the waters are calm so that tourists can enjoy the beautiful Pangandaran beach and engage in water sports on the beach. The west season occurs from November to April, where in this season tourist numbers are relatively lower due to sea conditions with large waves and relatively high rainfall, making it difficult for tourists to do water sports [9].

The quantitative descriptive method is used to describe the general condition of the research area, using primary and secondary data. The data were collected by interviewing 50 respondents drawn through accidental sampling framework. These respondents represent groups of tourists, public figures, and fishermen—all of whom have side jobs as marine ecotourism service providers. The analytical tool used to process the data is the Rapfish (rapid appraisal of fisheries) model approach.

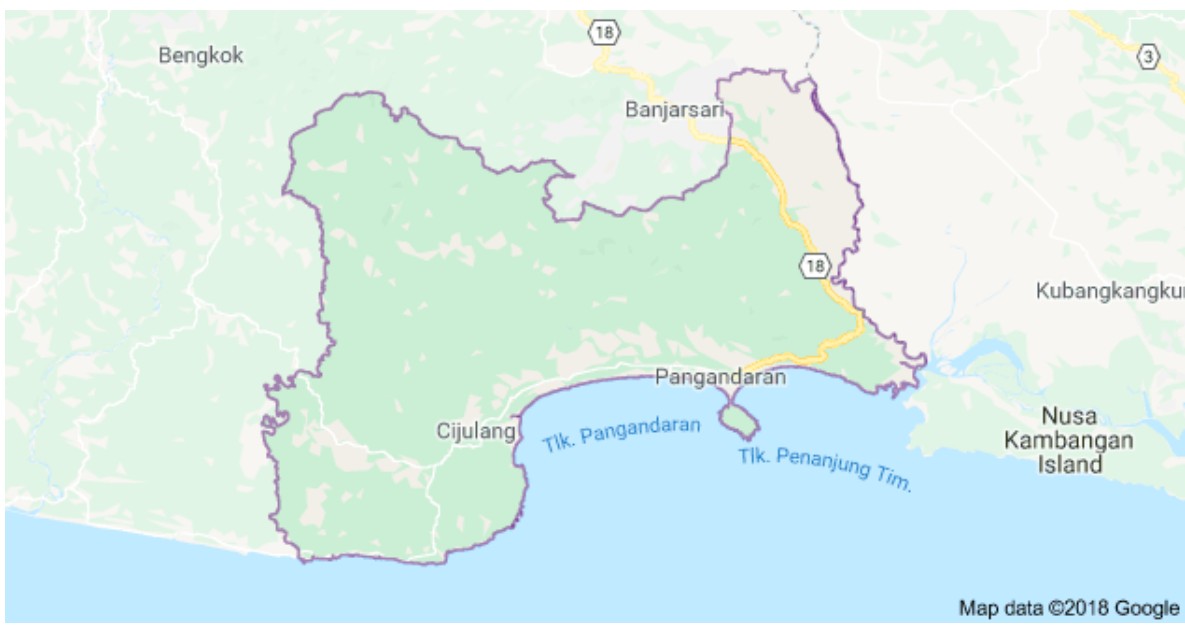

**Figure 1.** Map of Pangandaran, West Java Province, Indonesia.

The Rapfish framework adopts the multidimensional scaling (MDS) principles to assess the sustainability level of various marine ecotourism dimensions. This technique is basically a statistical calculation that performs a multidimensional transformation into more simple dimensions [10] to measure the synergistic model of sustainable development of marine ecotourism. In this research, five dimensions are assessed and these are environmental, cultural, social, economic, and infrastructural dimensions. Each dimension contains factors, called 'attributes' in this paper.

In the MDS, two points of the same object are mapped in far-flung points, which are very useful in regression analysis to calculate the "stress" that is a part of the MDS method [10–14]. Score on each attribute marine ecotourism will form a matrix X, where x is the number of areas and p is the number of attributes used. A good model is indicated by the S-stress value smaller than 0.25 or S < 0.25 and $R^2$ close to 1. Index scales that assess the sustainability of the system have an interval of 0–100%. In this research, there are four categories of status of marine ecotourism of sustainability, as seen in Table 1.

**Table 1.** Category index and status of sustainability for marine ecotourism.

| No | Index Value | Category |
|----|-------------|----------|
| 1. | 0.0–25.00 | Bad: not sustainable |
| 2 | 25.01–50.00 | Low: almost unsustainable |
| 3 | 50.01–75.00 | Sufficient: simply sustainable |
| 4 | 75.01–100.00 | Good: very sustainable |

The most important results of Rapfish analyses basically are: (1) scatter plot diagrams, (2) ordination graphs, and (3) leverage graphs. Scatter diagrams tell about an ordination technique that can produce a 'map' not far from the relative location. These maps can be rotated and shifted linearly with minimal disturbances. Group analysis of ordination points can be used to classify marine ecotourism mathematically.

Random marine ecotourism is represented by the cross in the center of the plot—it is normally distributed and the size of the arms of the cross is proportional to the 95% confidence limits on their standard errors. The cross has been displaced slightly vertically to make it visible: in fact, it lies almost precisely at the center of the ordination. The ordination graphs tell the location of sustainability indexes of the compared marine ecotourism for particular dimensions within a 'bad-to-good' scale. Meanwhile,

the leverage graphs indicate which attributes are most sensitive in a particular dimension; the longer the bar in the graph, the more sensitive the attribute is.

## 3. Results

The Pangandaran District is bordered by Ciamis in the North, Tasikmalaya in the West, Cilacap in the East, and the Indian Ocean in the South [14]. The coastal area surrounding this district belongs to six sub-districts. One of the primary purposes of visitors visiting the tourism centers in this district is to enjoy coral reef ecosystems as objects in diving and snorkeling activities. However, it is not only limited to enjoying the hard coral but also soft coral and other side events [7–9].

Based on the research, ecological dimensions determine significantly the suitability of marine ecotourism diving tourism categories, namely the brightness of the waters, coral community cover, type of life form, types of reef fish, current velocity, and depth of coral reefs. Whatever it is, the ecotourism activity interacts not only with ecological dimensions but also with those of social and economic systems. Therefore, the sustainability of of such a natural resource based ecotourism will depend also on a set of social and economic dimensions. The following are the result of Rapffish Analysis applied to measure the sustainability of marine ecotourism Pangandaran District. The results cover scatter plot, ordination, and leverage analysis, wherein five dimensions are considered: (1) environmental, (2) culture, (3) social, (4) economic, and (5) infrastructure.

### 3.1. Scatter Plotting/Monte Carlo Simulation

Monte Carlo simulation is essentially intended to see the level of disturbance (perturbation) to the value of the ordinance [14] and carried out by iteration 25 times. The results of Monte Carlo analysis through scatter plots in the environmental dimension have experienced disturbance that will threaten the sustainability of marine ecotourism in Pangandaran region. The results can be seen in Figures 2–6. As seen in these figures, it can be concluded that the data used in this analysis are reliable.

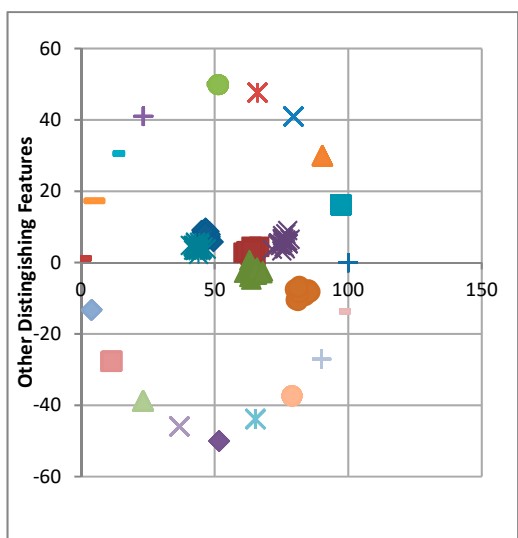

**Figure 2.** Scatter plot for the enviromental dimension.

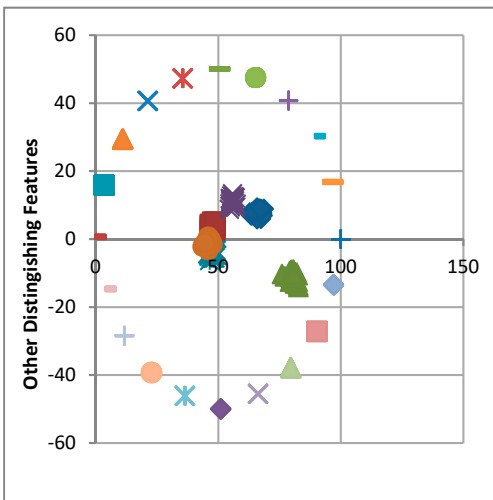

**Figure 3.** Scatter plot for the cultural dimension.

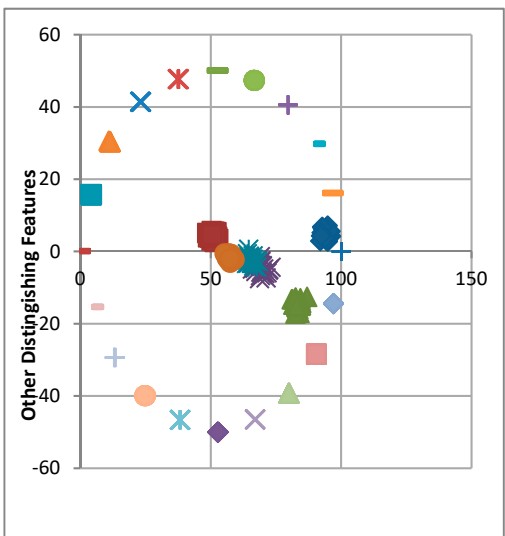

**Figure 4.** Scatter plot for the economic dimension.

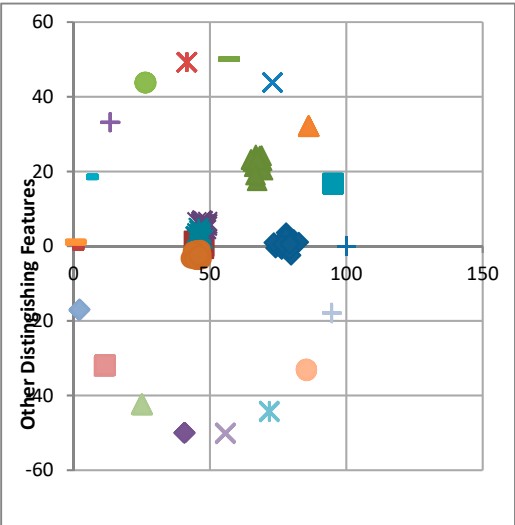

**Figure 5.** Scatter plot for the infrastructural dimension.

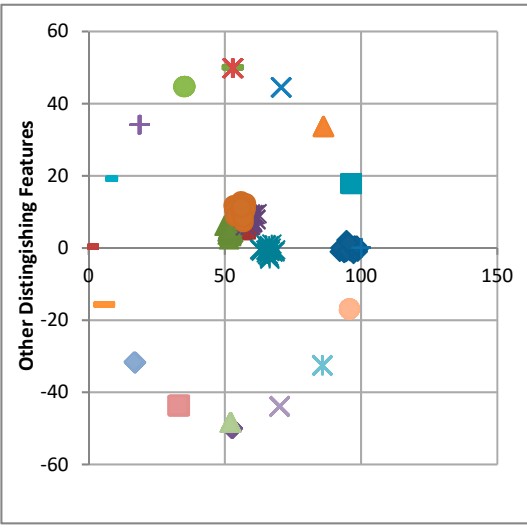

**Figure 6.** Scatter plot for the social dimension.

### 3.2. Ordination

In Figures 7–11 the horizontal axis shows the difference in type of marine ecotourism in bad (0%) to good (100%) ordinations for each dimension analyzed, while the vertical axis shows the difference from the attribute mix score between the type of marine ecotourism evaluated. The ordination analysis shows that the sustainability of marine ecotourism in the Pangandaran region varies between type of marine ecotourism. Below are graphs showing loci of sustainability indexes of each type of ecotourism based on a five-dimension point of view.

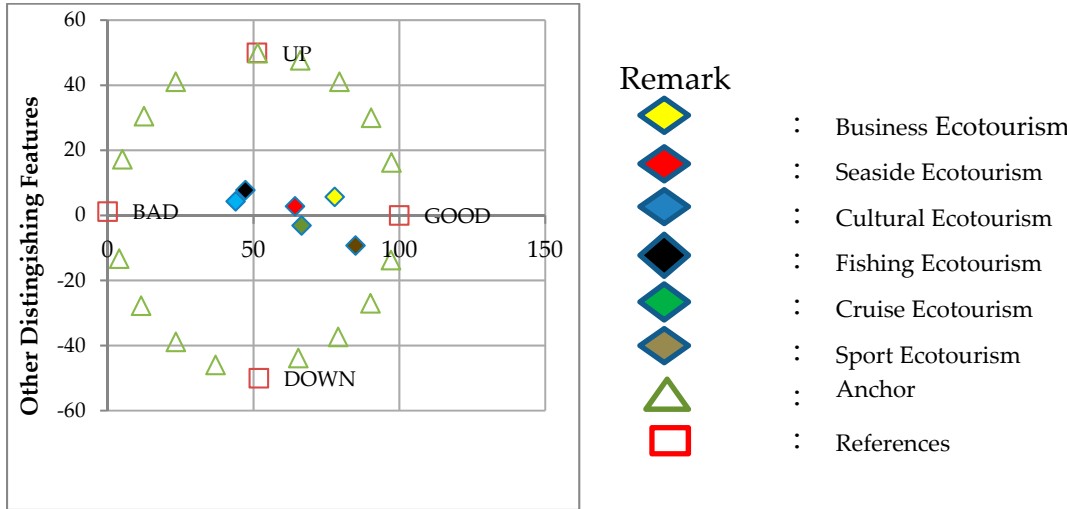

**Figure 7.** Rapfish ordination of the environmental dimension.

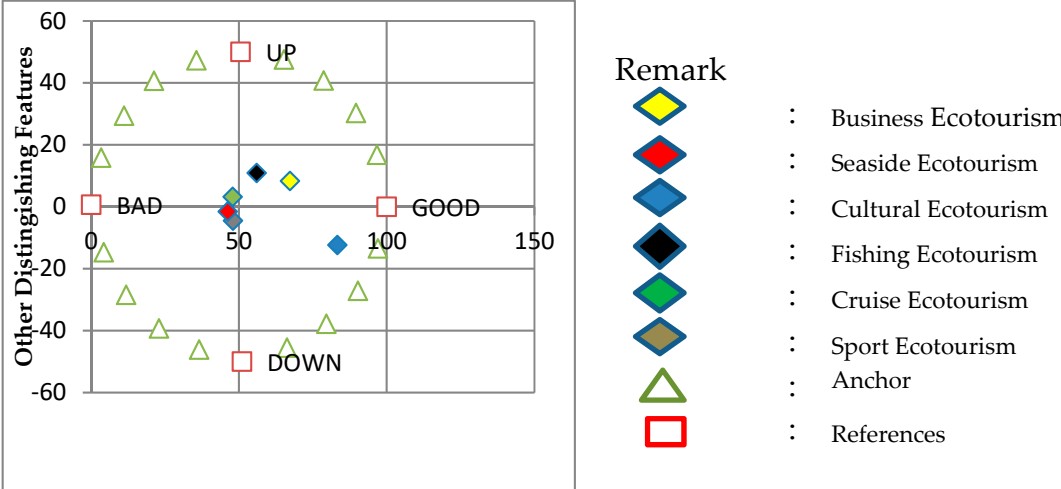

**Figure 8.** Rapfish ordination of the cultural dimension.

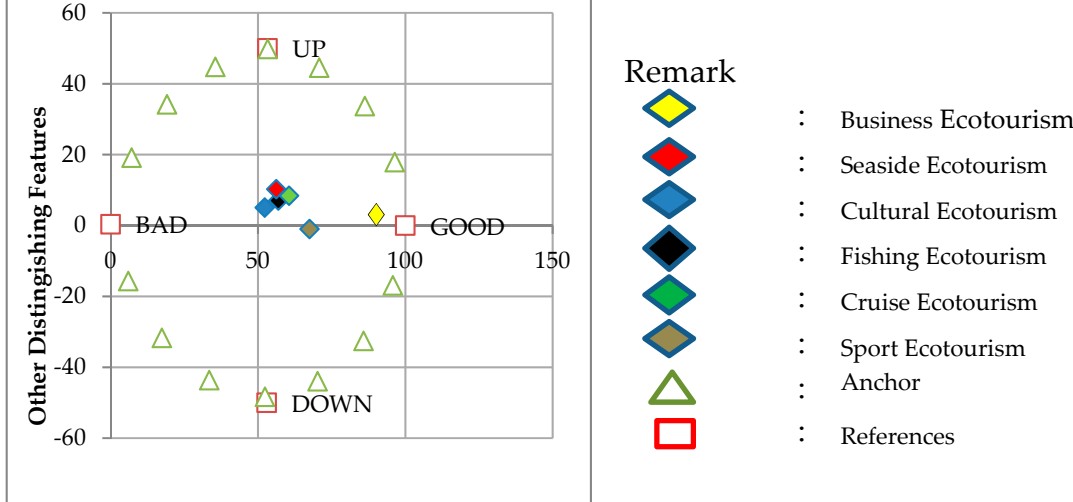

**Figure 9.** Rapfish ordination of the social dimension.

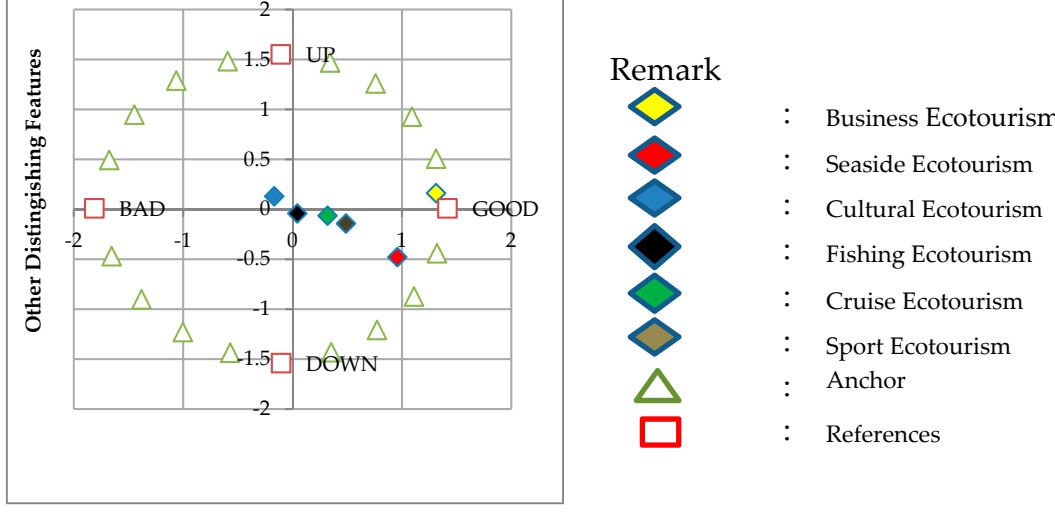

**Figure 10.** Rapfish ordination of the economic dimension.

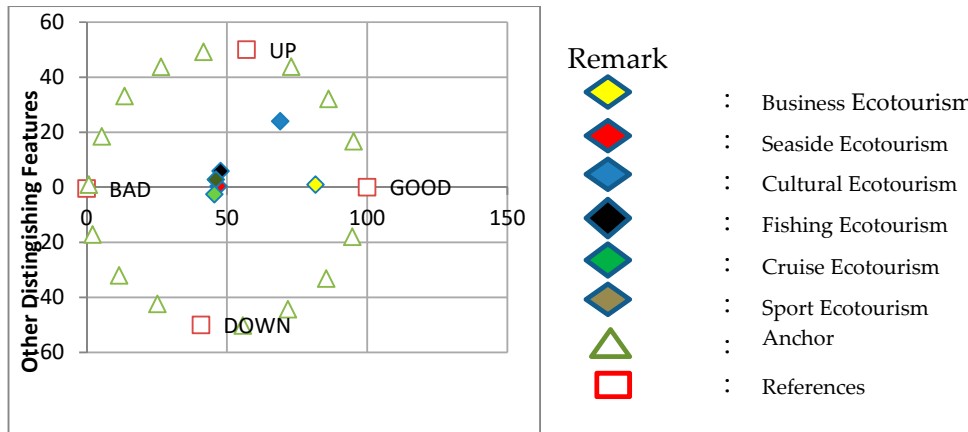

**Figure 11.** Rapfish ordination of the infrastructural dimension.

Figure 7 shows the ordinance analysis in the enviromental dimension with two time iterations resulting in a quadratic value of correlation ($R^2$) of 93.73% and stress value (S) of 17.18%. From this stability indicator, it can be seen how far the results of the analysis are reliable. Thus, the analysis of the environmental dimension in this research shows the condition of goodness of fit, considering the stress value obtained is 17.18%. (<25%). Then we divide the scale of the ordinance into four groups with different levels of sustainability, namely 0–25 is bad; 26–50 is low; 51–75 is sufficient; and 76–100 is good. The result can be seen in Table 2 as follows.

**Table 2.** Sustainability level of marine ecotourism environmental dimensions in the Pangandaran Region.

| No | Type of Marine Ecotourism | Dimension Enviromental | Status of Sustainability |
|----|---------------------------|------------------------|--------------------------|
| 1. | Business Ecotourism | 47.119 | Low |
| 2. | Seasides Ecotourism | 64.306 | Sufficient |
| 3. | Cultural Ecotourism | 66.298 | Sufficient |
| 4. | Fishing Ecotourism | 77.999 | Good |
| 5. | Cruise Ecotourism | 43.796 | Less |
| 6. | Sport Ecotourism | 85.186 | Good |

Figure 8 shows the ordinance analysis in the cultural dimension with two time iterations results in a quadratic value of correlation ($R^2$) of 93.50% and stress value (S) of 18.68%. From this stability indicator, it can be seen how far the results of the analysis are reliable. Thus, the analysis of cultural dimensions in this research shows the condition of goodnes of fit, considering the value of stress obtained is equal to 18.68%. (<25%). Thus, the analysis of cultural dimensions in this research shows the condition of goodnes of fit, considering the value of stress obtained is equal to 18.68%. (<25%). Then we divide the scale of the ordinance into four groups with different levels of sustainability, namely 0–25 is bad; 26–50 is low; 51–75 is sufficient and 76–100 is good. The result can be seen in Table 3 as follows:

**Table 3.** Sustainability level of types of marine ecotourism cultural dimensions in the Pangandaran Region.

| No | Type of Marine Ecotourism | Dimension Cultural | Status of Sustainability |
|----|---------------------------|--------------------|--------------------------|
| 1. | Business Ecotourism | 67.259 | Suffcient |
| 2. | Seasides Ecotourism | 47.820 | Less |
| 3. | Cultural Ecotourism | 83.253 | Good |
| 4. | Fishing Ecotourism | 55.973 | Suffcient |
| 5. | Cruise Ecotourism | 47.971 | Less |
| 6. | Sport Ecotourism | 46.158 | Less |

Figure 9 shows the ordination analysis in terms of social dimensions between good and bad. Ordinance analysis in the social dimension with the number of iterations is two times, resulting in a quadratic value of correlation ($R^2$) of 92.62% and stress value (S) of 18.81%. Thus, the analysis of the social dimension in this research shows the condition of goodness of fit, considering the stress value obtained is 18.81%. (<25%). Then we divide the scale of the ordinance into four groups with different levels of sustainability, namely 0–25 is bad; 26–50 is low; 51–75 is sufficient; and 76–100 is good. The result can be seen in Table 4 as follows.

**Table 4.** Sustainability level type of marine ecotourism social dimension in Pangandaran Region.

| No | Type of Marine Ecotourism | Cultural Dimension | Status of Sustainability |
|---|---|---|---|
| 1. | Business Ecotourism | 99.903 | Good |
| 2. | Seasides Ecotourism | 56.861 | Sufficient |
| 3. | Cultural Ecotourism | 52.295 | Sufficient |
| 4. | Fishing Ecotourism | 60.510 | Sufficient |
| 5. | Cruise Ecotourism | 67.461 | Sufficient |
| 6. | Sport Ecotourism | 56.177 | Sufficient |

Figure 10 shows the ordinance analysis in the economic dimension with two time iterations results in a quadratic value of correlation ($R^2$) of 94.64% and stress value (S) of 17.21%. Thus, the economic dimension analysis in this reserach shows the condition of goodnes of fit, considering the value of stress obtained is 17.21%. (<25%). Then we divide the scale of the ordinance into four groups with different levels of sustainability, namely 0–25 is bad; 26–50 is low; 51–75 is sufficient; and 76–100 is good. The result can be seen in Table 5 as follows.

**Table 5.** Sustainability level type of marine ecotourism economic dimension in Pangandaran Region.

| No | Type of Marine Ecotourism | Economic Dimension | Status of Sustainability |
|---|---|---|---|
| 1. | Business Ecotourism | 96.754 | Good |
| 2. | Seasides Ecotourism | 50.948 | Sufficient |
| 3. | Cultural Ecotourism | 85.784 | Sufficient |
| 4. | Fishing Ecotourism | 71.250 | Sufficent |
| 5. | Cruise Ecotourism | 66.048 | Sufficient |
| 6. | Sport Ecotourism | 57.453 | Sufficient |

Figure 11 shows the ordinance analysis in the infrastructure dimension with two time iterations resulting in a quadratic value of correlation ($R^2$) of 93.02% and stress value (S) of 17.42%. Thus, the infrastructure dimension analysis in this reserach shows the condition of goodnes of fit, considering the value of stress obtained is 15.21% (<25%). Then we divide the scale of the ordinance into four groups with different levels of sustainability, namely 0–25 is bad; 26–50 is low; 51–75 is sufficient; and 76–100 is good. The result can be seen in Table 6 as follows.

**Table 6.** Sustainability level type of marine ecotourism infrastructure dimension in Pangandaran Region.

| No | Type of Marine Ecotourism | Dimension Economic | Status of Sustainability |
|---|---|---|---|
| 1. | Business Ecotourism | 81.529 | Good |
| 2. | Seasides Ecotourism | 46.992 | Low |
| 3. | Cultural Ecotourism | 68.998 | Sufficient |
| 4. | Fishing Ecotourism | 47.697 | Low |
| 5. | Cruise Ecotourism | 46.022 | Low |
| 6. | Sport Ecotourism | 45.510 | Low |

## 4. Discussion

Ecotourism potentially provides a sustainable approach to development [15]. In this scope, marine ecotourism is a form of natural marine resource-based tourism that is educational, low-impact, non-consumptive, and locally oriented: local people must control the industry and receive the bulk of the benefits to ensure sustainable development [16]. Ecotourism in this context can be viewed as an activity to promote responsible travel to natural areas, to make a positive contribution to environmental preservation, and to improve the welfare of local communities [17–19].

### 4.1. Environmental Dimension of Marine Ecotourism

The elements of the environmental dimension of marine ecotourism include: (1) nature conservation, (2) natural value, (3) protected are of nature, (4) disrupting wildlife, (5) illegal hunting and fishing, (6) degradation water quality, (7) disruption of local flora and fauna, (8) bidiversity loss, (9) habitat alteration, and (10) environmental education.

In Figure 12, it can be seen that the highest value of 3.171 belongs to the 'illegal hunting and fishing' attribute, which means that this attribute has the highest sensitivity value related to the level of marine ecotourism environmental dimension sustainability. The 'environmental education' attribute has the lowest value of 0.808, meaning that it has little sensitivity to the level of sustainability of marine tourism.

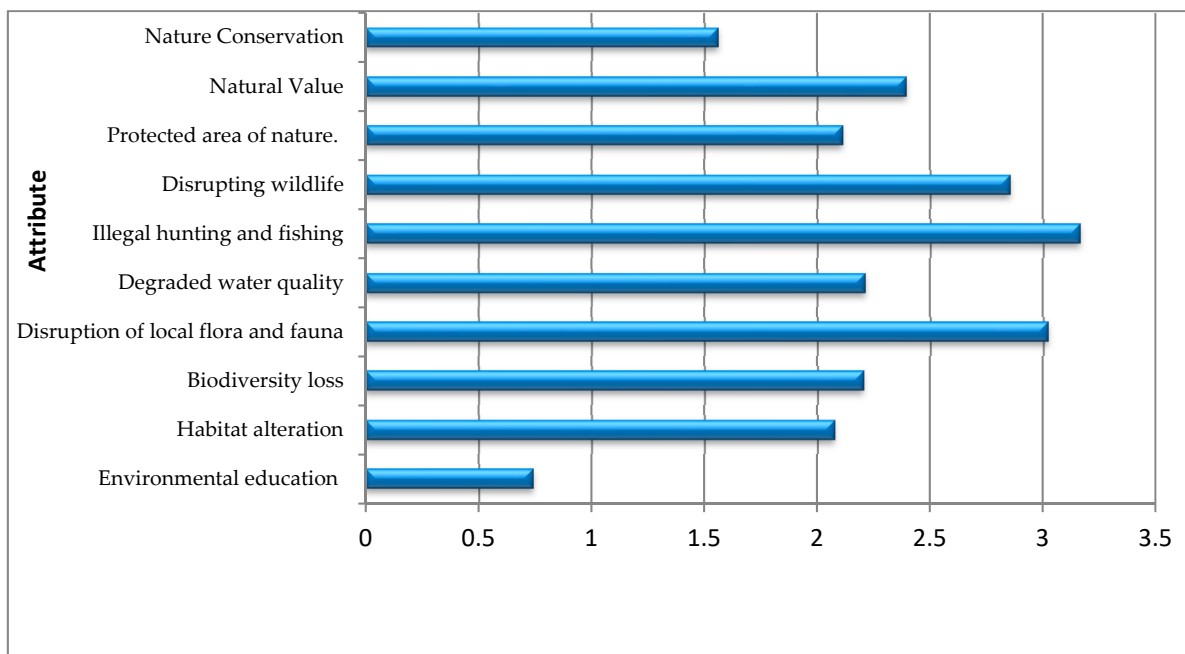

**Figure 12.** Leverage of elements of the enviromental dimension of marine ecoturism on a sustainability scale of 0 to 100.

### 4.2. Cultural Dimension of Marine Ecotourism

The culture of coastal communities is different from other communities, humans are cultural beings, and culture is the result of creativity, work, and joint initiatives. One of the factors that influence the formation of culture is the physical, natural environment; such situations and conditions will indirectly shape the character of the personality and culture of the people who live in that environment. The dependence of the community on the marine sector provides its own identity as a coastal community with a lifestyle known as 'coastal culture' [15].

The elelments of the cultural dimension of marine ecotourism include: (1) creating sustainable livelihoods; (2) traditional ethnic; (3) behavioral patterns; (4) religious beliefs; (5) existing skill levels;

(5) cultural attractions; (6) practicing respect for local culture; (7) local and national heritage; (8) indiegenous culture; and (9) adaptation to local norms. In Figure 13, it can be seen that the highest value of 3.068 belongs to the 'existing skill' attribute, and thus it is the most sensitive among the cultural dimensions of marine ecotourism. The 'traditional ethnicity' attribute has the lowest value of 0.701, meaning that it has little sensitivity to the level of sustainability of marine tourism.

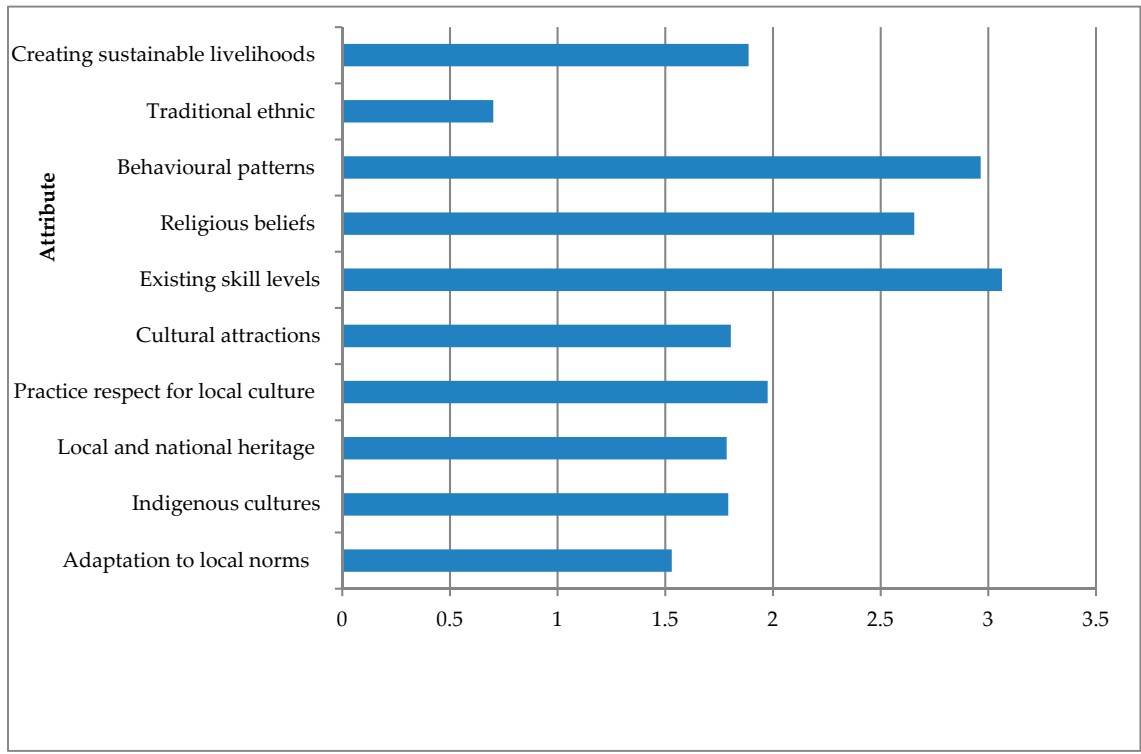

**Figure 13.** Leverage of elements of the cultural dimension of marine ecoturism on a sustainability scale 0 to 100.

### 4.3. Social Dimension of Marine Ecotourism

The social dimension is a person's actions in certain ways in an effort to exercise rights and obligations in accordance with status they have. A person can be said to play a role if he had carried out their rights and obligations in accordance with their social status within society. The World Tourism Organization (WTO), states that: "Tourism comprises the activities of persons, traveling to and staying in place outside their usual environment for not more than one consecutive year for leisure, business and other purposes" [16].

The elements of the social dimension of marine ecotourism include (1) ecotourism income; (2) benefit for local people; (3) conflict status; (4) education level of tourism; (5) number of tourists; (6) type of tourists; (7) traditional events; and (8) enforcement of regulations. In Figure 14, it can be seen that the highest value of 3.660 belongs to 'number of tourists', and it means that it has the greatest sensitivity with respect to the level of social dimension sustainability of marine ecotourism compared to other attributes. The 'education level' attribute of tourism has the lowest value of 1.239, meaning that it is the least sensitive in terms of its influence on the level of sustainability of this particular marine tourism dimension.

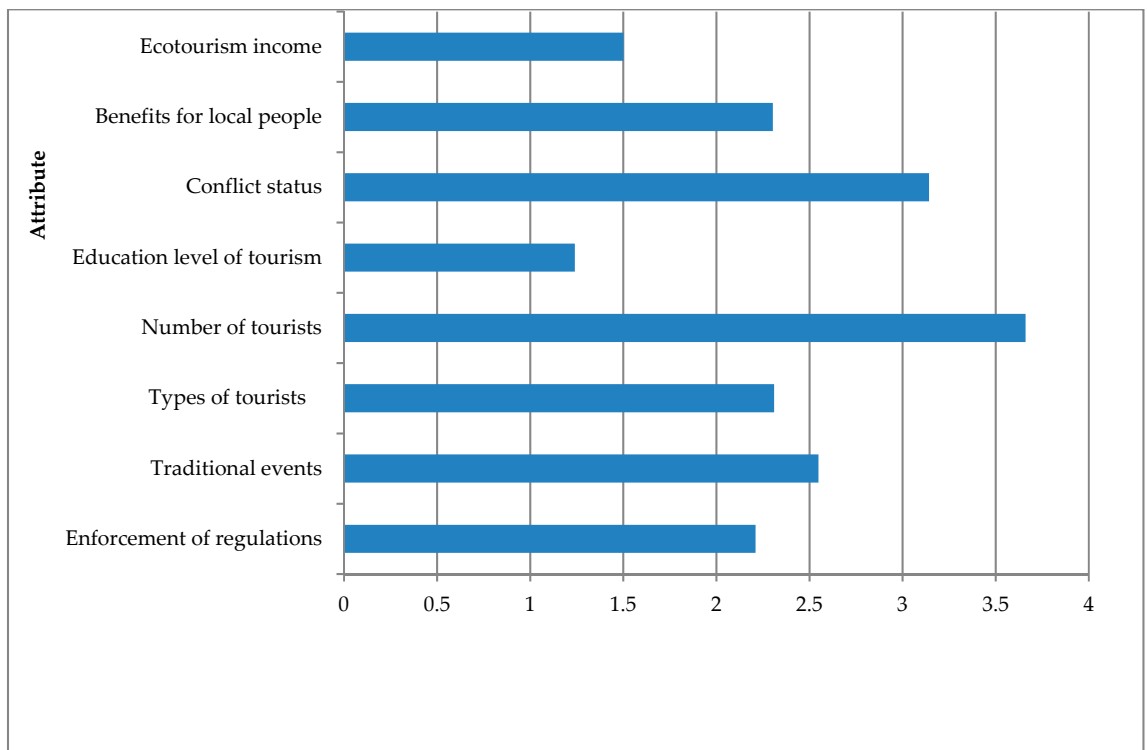

**Figure 14.** Leverage of elements of the social dimension of marine ecoturism on a sustainability scale of 0 to 100.

### 4.4. Economic Dimension of Marine Ecotourism

The elements of the economic dimension of marine ecotourism include (1) domestic ecotourism investors; (2) foreign ecotourism investors; (3) ecotourism industry; (4) jobs for local communities; (5) other income; (6) marketing techniques; (7) employment in ecotourism; (8) average wage; (9) ecotourism entrepreneurship; and (10) providing benefits for local communities.

In Figure 15, it can be seen that the highest value of 2.953 belongs to the foreign ecotourism investors attribute, so it means that the foreign ecotourism investors attribute has the greatest sensitivity with respect to the level of sustainability of the economic dimension of marine ecotourism.

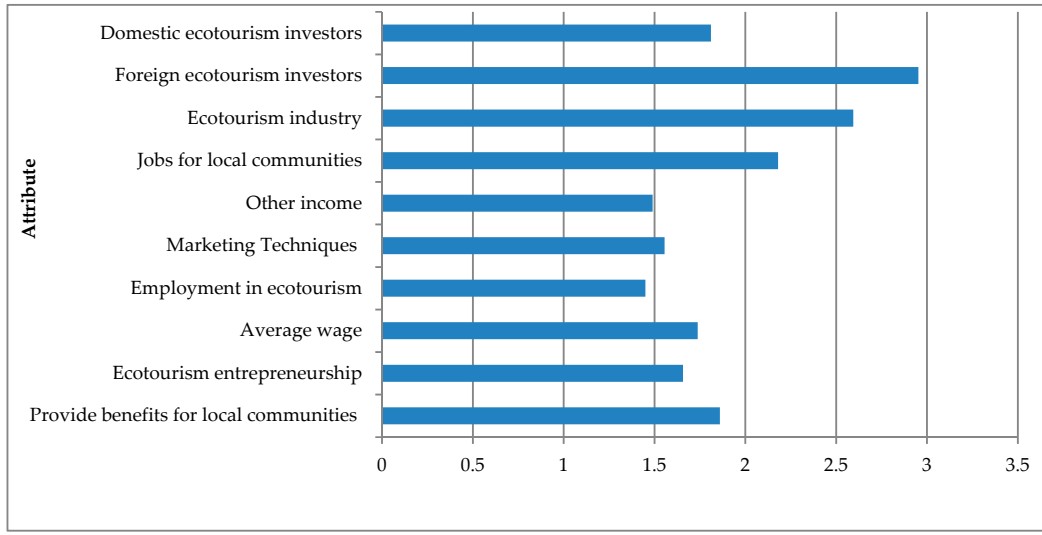

**Figure 15.** Leverage of elements of the economic dimension of marine ecoturism on a sustainability scale of 0 to 100.

*4.5. Infrastructural Dimension of Marine Ecotourism*

Elements of theinfrastructural dimension of marine ecotourism include (1) lodging, (2) tourism support service, (3) restaurants and markets, (4) fuel, (5) health care and service, (6) public administration, (7) communication servce, (8) new sport recreational, and (9) transportation.

In Figure 16, it can be seen that the highest value of 4.149 belongs to health care service, which means that it has the highest sensitivity value with respect to the level of sustainability of the infrastructural dimension of marine ecotourism. Tourism support services has the lowest value of 1.444. Tourism support service, therefore, it has the smallest value of sensitivity to this dimension of sustainability.

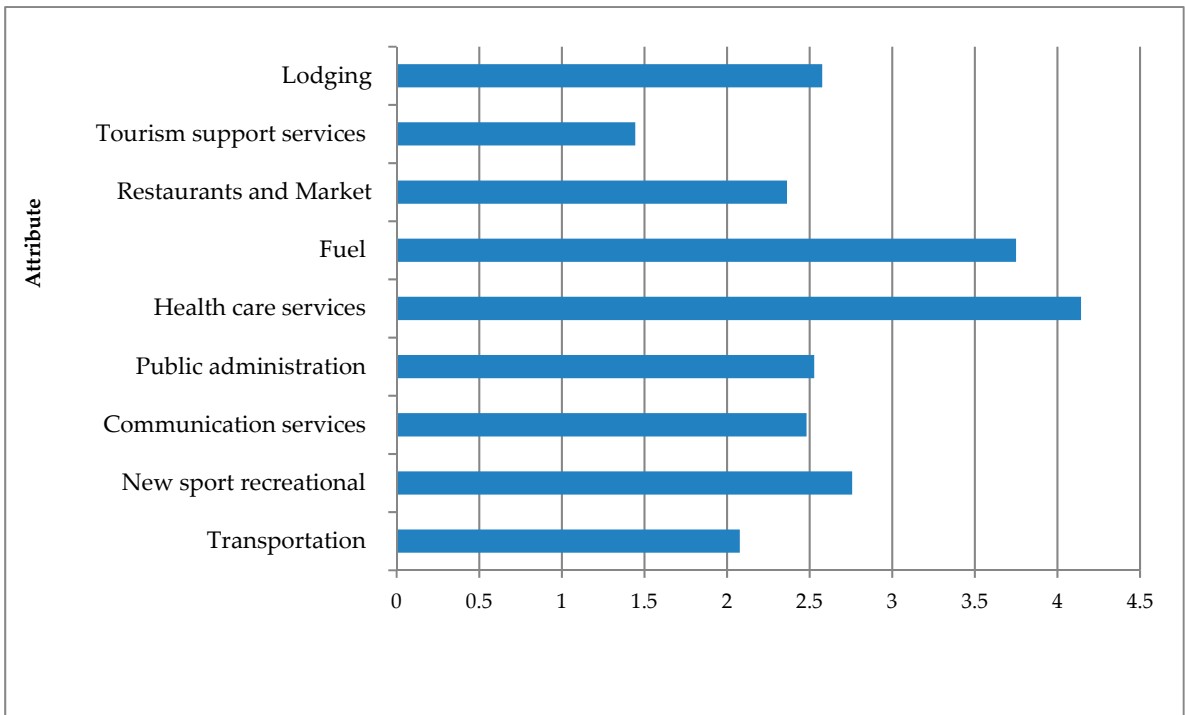

**Figure 16.** Leverange of elements of the infrastructural dimension of marine rcotourism on a sustainability scale of 0 to 100.

Based on this research, the Rapfish model measured the synergistic model of sustainable development of marine ecotourism through the approach environment, culture, social, economic and infrastructure dimension. The sustainability levels by type of environmental dimensions of marine ecotourism in Pangandaran region were found to be as follows: (1) business ecotourism is low; (2) seaside ecotourism is sufficient; (3) cultural ecotourism is sufficient; (4) fishing ecotourism is good; (5) cruise ecotourism is low; and (6) sport ecotourism is good. Marine ecotourism business is very complex, requiring entrepreneurial spirit to achieve profitability with no damage to the environment. Start-up ecotourism ventures have a high risk of failure and the marine tourism business faces challenges in conditions of uncertainty in natural resources. Environmental dimensions include: (1) nature conservation, (2) natural value, (3) protected are of nature, (4) disrupting wildlife, (5) illegal hunting and fishing, (6) degradation water quality, (7) disruption of local flora and fauna, (8) biodiversity loss, (9) habitat alteration, and (10) environmental education.

Maritime ecotourism focuses on local cultures from certain areas, including coastal areas, as well as natural beauty, geological structures, natural vegetation, and fauna [20,21] and is a type of tourism that covers the subject of conservation of natural areas, education, economic benefits, quality tourism, and local community participation [22]. Based on this research, sustainability levels of types of marine ecotourism cultural dimension in Pangandaran Region were found to be as follows: (1) business

ecotourism is sufficient; (2) seasides ecotourism is low; (3) cultural ecotourism is good; (4) fishing ecotourism is sufficent; (5) cruise ecotourism is low; and (6) sport ecotourism is low. There are three main principles in sustainable development [23]: (1) ecological sustainability, namely ensuring that development is carried out in accordance with ecological, biological, and diversity of existing ecological resources; (2) social and cultural sustainability, namely ensuring that the development carried out has a positive impact on the lives of the surrounding community and in accordance with the culture and values that apply to the community; (3) economic sustainability, namely ensuring that development is carried out efficiently economically and that the resources used can survive for future needs. Based on this research, cultural dimensions include: (1) creating sustainable livelihoods; (2) traditional ethnic; (3) behavioral patterns; (4) religious beliefs; (5) existing skill levels; (5) cultural attractions; (6) practising respect for local culture; (7) local and national heritage; (8) indegenous culture; and (9) adaptation to local norms.

From a sociological perspective, marine ecotourism systems have three types of actors: (1) tourism brokers, (2) local tourism residents, and (3) tourists [24]. Interactions within and between these actors can affect the speed and character of coastal development and increase the income of coastal communities. Based on this research, the sustainability level of types of marine ecotourism social dimension in Pangandaran Region are as follows: (1) business ecotourism is good; (2) seaside ecotourism is sufficient; (3) cultural ecotourism is sufficient; (4) fishing ecotourism is sufficient; (5) cruise ecotourism is sufficient; and (6) sport ecotourism is sufficient. Maritime tourism not only promotes local economic growth, but also promotes social equality rights in the community and preserves the surrounding environment. Social dimensions include (1) ecotourism income; (2) benefits for local people; (3) conflict status; (4) education level of tourism; (5) number of tourists; (6) type of tourists; (7) traditional events; and (8) enforcement of regulations.

Tourism is considered as combining time and pleasure, benefiting prospective tourists, and providing the tourism industry and host countries with significant flowing effects at all levels and sectors in the local economy [25]. Based on this research, sustainability levels according to the type of economic dimension of marine ecotourism in Pangandaran Region are as follows: (1) business ecotourism is good; (2) seasides ecotourism is sufficient; (3) cultural ecotourism is sufficient; (4) fishing ecotourism is sufficient; (5) cruise ecotourism is sufficient; (6) sport ecotourism is sufficient. Economic dimensions include; (1) domestic ecotourism investors; (2) foreign ecotourism investors; (3) ecotourism industry; (4) jobs for local communities; (5) other income; (6) marketing techniques; (7) employment in ecotourism; (8) average wage; (9) ecotourism entrepreneurship; and (10) providing benefits for local communities.

Based on this research, sustainability levels for the infrastructural dimensions of marine ecotourism in the Pangandaran Region are as follows: (1) business ecotourism is good; (2) seaside ecotourism is sufficient; (3) cultural ecotourism is sufficient; (4) fishing ecotourism is sufficient; (5) cruise ecotourism is sufficient; and (6) sport ecotourism is sufficient. Infrastructural dimensions include: (1) lodging; (2) tourism support services; (3) restaurants and markets; (4) fuel; (5) health care service; (6) public administration; (7) communication service; (8) new sport recreational; and (9) transportation.

## 5. Conclusions

A development model for synergistic sustainable marine ecotourism (Case Study in Pangandaran Region, West Java Province) through a multidimensional scaling approach was presented. The dimesions involved are environmental, cultural, social, economic, and infrastructural dimensions. These dimensions demonstrate sufficient conditions to support the sustainability of marine ecotourism in the Pangandaran región. However, coastal natural resources in the Pangandaran area still need to be improved and maintained through good management. Strategies that can be employed to increase the number of tourists include increasing access to transportation, information, and adequate accommodations in accordance with tourism standards.

**Author Contributions:** Conceptualization, A.N. and A.K.S.; Data curation and formal analysis, A.N, I.A., and A.K.S.; Funding acquisition, A.K.S.; Methodology, A.N., I.A., and A.K.S.; Resources, A.N.; Software, A.N. and A.K.S.; Visualization, A.N.; Writing—original draft, A.N. and A.K.S. Writing—review and editing, A.N.

**Funding:** This research was funded by the Academic Leadership Grant (ALG-2019) Universitas Padjadjaran, Bandung, Indonesia.

**Conflicts of Interest:** The authors declare no conflict of interest.

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
