# Peer review of "Model Development of A Synergistic Sustainable Marine Ecotourism—A Case Study in Pangandaran Region, West Java Province, Indonesia"

_sustainability, doi:10.3390/su11123418_

Round 1
Reviewer 1 Report
This is now the third time I have reviewed this manuscript. This review has not taken more than 2 minutes, I simply looked at figures 2-5 and again, they are identical.
This shouldn't be the case, and something I have commented on both times before. This gives me no confidence in the analysis and I can't see why this point is not addressed by the authors (there seems to be no response to reviewer comments).
Author Response
The figures have been revised. Earlier version contains duplicated figures (Figs, 2-5). These have been rectified by re-run the program to produce appropriate figures for each cases. Thank you to the reviewer for rising this concern.

Reviewer 2 Report
The paper "Development Model of Synergistic Sustainable Marine Ecotourism (Case Study in Pangandaran Region, West Java Province,Indonesia) use the Rapfish analysis tool as a model to approach the measuring of synergistic model of sustainable development of marine ecotourism. I had already reviewed this paper valued it ready for pubblication in present form.
I confirm yet my previous evalutaion
Author Response
Thank you for the positive comments.

Reviewer 3 Report
Dear colleagues,
Your research paper is quite sound and properly presented. There some minor points you should attend a bit:
Line 77: "This research aims to analysis development modelof synergistic sustainable marine ecotourism" might be put as "This research aims to analyze (or, aims at the analysis of) thedevelopment model of synergistic sustainable marine ecotourism"
Line 145: " it can be cocludedthat…" should be " it can be concludedthat…"
Lines 210-212: "
"Ecotourism to promote responsible travel to natural areas, to make a positive contribution to environmental preservation and to improve the welfare of local communities" should be rephrased to make the utterance syntactically correct.
Line 214: "Attributeof environmental dimensions marine ecoturism include:" should be put as "The Attribute (The attributes)of environmental dimensions marine ecotourism includes (include):"
As well the term "ecotourism" is to be checked in the table below in terms of spelling
Author Response
All the comments have been included in the revised paper. Thank you very much for the constructive comments.

This manuscript is a resubmission of an earlier submission. The following is a list of the peer review reports and author responses from that submission.
Round 1
Reviewer 1 Report
The paper titled "Development Model of Synergistic Sustainable Marine Ecotourism (Case Study in Pangandaran Region, West Java Province,Indonesia), it's an interesting point of view on the evaluation of sustainability of coastal turism. The authors have had used Rapfish technique approach a current issue for many coastal areas.
The authors have had used Rapfish technique applying the MDS principles.
I wen reading M&M and the subsequent paragraphs on the result and i found it very unclear and difficult to read by the most readers of "Sustainabilty".
My personal suggestion for the authors is: rewrite the M&M part explaining better the Rapfish/MDS and how the graphs and plots shoud be read.
For results paragraph:
i suggest to resume the parts on the five "dimansion"
and rearrange the plots in a page single page, adding the figure captions and the plots legend. If possible use colored figures.
In your research graphical results are very important to explane results and to discuss it.
In opinion of this reviewer, major revision are suggested to improving M&M presentation.
Reviewer 2 Report
The subject of this paper is interesting, and I would encourage resubmission of the results once some major issues with the data have been resolved. I can't really follow the methods here - what factors are put into the analysis? I'm not sure this is clear, or appropriately done, as as far as I can see, figures 1, 5, 9, 13 and 17 are identical. Figures 2, 6, 10, 14 and 18 are identical and figures 3, 7, 11, 15, 19 are identical.
Once clarification on the results and methods have been made, and appropriate changes to results and discussion, I would encourage resubmission of the manuscript
Round 2
Reviewer 1 Report
The authors of paper titled "Development Model of Synergistic Sustainable Marine Ecotourism " have follow the reviewer suggestions to improve the general quality of the paper. In opinion of this reviewer, after the last revision, the paper are been ready for pubbication.
Reviewer 2 Report
I'm still confused by this paper and the analysis in it. I'm afraid I don't think it is correct at present. Figures 2-5 are now identical, but surely this shouldn't be the case? There is little detail on what these figures mean or how they have been produced, so I can't offer any more guidance on this. I can't see how the authors altered the previous version - the response is just a copy and paste from the original and new manuscript.
I'm afraid until the methods are transparent, and there are no obvious mistakes such as identical figures for different dependent variables in the results, I don't think this paper is suitable for publication.